# Honey Bioactive Molecules: There Is a World Beyond the Sugars

**DOI:** 10.3390/biotech13040047

**Published:** 2024-11-14

**Authors:** Gregorio Bonsignore, Simona Martinotti, Elia Ranzato

**Affiliations:** Dipartimento di Scienze e Innovazione Tecnologica (DiSIT), University of Piemonte Orientale, 15121 Alessandria, Italy; gregorio.bonsignore@uniupo.it (G.B.); simona.martinotti@uniupo.it (S.M.)

**Keywords:** antioxidant, anti-inflammatory, immunomodulatory, honey, polyphenols, antibacterial activity, wound repair

## Abstract

Honey’s many bioactive compounds have been utilized historically to cure infectious diseases. Beneficial effects are its antiviral, antibacterial, anti-inflammatory, antioxidant, and immune-stimulating qualities. The bee species, geographic location, botanical origin, harvest season, processing, and storage conditions all affect honey’s potential for therapeutic use. Honey contains a number of antioxidants and active compounds, such as polyphenols, which have been shown to have disease-preventive properties. Based on their origins, categories, and functions, the main polyphenols found in various honey varieties are examined in this review.

## 1. Introduction

Since ancient times, honey has been used traditionally to treat infectious ailments [1]. Because so many different bioactive chemicals are involved, honey’s extraordinary qualities are complex. Honey is steadily gaining credibility as a reliable and efficient therapeutic agent to both the general public and conventional medical professionals [2,3,4].

Honey’s antibacterial, anti-inflammatory, antiviral, antioxidant, and immune-stimulating properties have all been linked to its positive effects (Figure 1). [5]. Honey has been demonstrated to effectively cure a variety of other illnesses, such as arthritis, gastric ulcers, diarrhea, dermatitis, and diabetes, and for wound healing [6].

Honey can also be used to disinfect wounds and cleanse skin [7]. Honey’s high sugar content, acidity, hydrogen peroxide content, and other non-peroxide molecules are the majority of the known reasons that give honey these characteristics [8,9,10,11].

Researchers have just rediscovered honey’s inherent antibacterial qualities. The growing trend of antibiotic resistance and the lack of promising new treatments has stoked interest in honey’s potential antibacterial properties.

Honey has been shown in numerous studies to have antibacterial capabilities. Even though honey is produced all over the world, its medicinal qualities can differ and are primarily based on the bee species involved, its geographic region, and its botanical origin [12]. Harvesting season, processing, storage conditions, and environmental considerations are external features that might be involved.

Honey’s medicinal potential is quite complicated because of the action of different substances, in addition to the significant differences in these substances’ quantities between honeys. The medicinal qualities of honey may be impacted by a number of things, like heat and light exposure [13,14,15].

In fact, numerous active ingredients and antioxidants, including polyphenols, are found in honey. Phytochemicals, a general word for the thousands of plant-based compounds with antioxidant qualities, include polyphenols. The preventive effects of polyphenols against various diseases have been confirmed by several pieces of animal research and numerous in vitro and in vivo experiments. However, it is difficult to pinpoint the precise biological process that underlies each polyphenol and to ascertain the effects of polyphenols on human health [16].

In this review, we explore the most abundant polyphenols found in different kinds of honey based on their source, classification, and specialized roles.

## 2. Honey and Humankind

Natural honey has the highest energy density of any food. Therefore, it should come as no surprise that honey is a staple food for practically all hunter-gatherers in areas where it is found.

There is also evidence from Stone Age drawings suggesting that honeybee remedies for ailments date back more than 8000 years. Historical documents including old books, tablets, and parchments have documented the use of honey as medicine: Sumerian chalcolithic tablets (6200 BC), the 5000-year-old Veda (a sacred scripture in Hinduism), the Holy Bible, the Koran, and Hippocrates (460–357 BC), as well as Egyptian papyri (1900–1250 BC) [17,18].

### 2.1. Honey as an Antimicrobial Agent

Honey has demonstrated its efficaciousness as an antibacterial agent against both pathogenic and non-pathogenic microorganisms, including bacteria, yeasts, and fungi, even against those that have evolved a resistance to a variety of antibiotics. Honey has antimicrobial properties that can be either bacteriostatic or bactericidal, depending on the dosage [13,19]. Despite this, several factors, including high osmolarity (low water action), low pH (acidity), hydrogen peroxide (H_2_O_2_), and non-peroxide components have been linked to its potentials [20,21,22].

Furthermore, supersaturated sugars, which have a strong affinity for water molecules, are present in bee honey. As a result, bacteria eventually become dehydrated and perish [19].

According to Fahim and coworkers, honey’s natural acidity inhibits the growth of microorganisms [23]. Nevertheless, the primary antibacterial activity has been attributed to hydrogen peroxide activity, which is a by-product of the glucose-oxidase enzyme oxidizing glucose, particularly in diluted honey form. Hydrogen peroxide breaks down into very reactive free radicals, which react by killing microorganisms. In general, the honey’s ability to retain its properties can be readily interrupted by heat or catalase activity [24].

Honey’s antibacterial activity may not be due to peroxides, but rather to non-peroxides molecules, resulting in more consistent and long-lasting benefits.

The oligosaccharides found in honey have additional significant impacts. They have many of the same prebiotic characteristics as fructo-oligosaccharides. Some data have indicated that oligosaccharides are responsible for the increase in population of some advantageous microorganisms responsible for maintaining a healthy gut microbiota in humans, such as *Lactobacilli* and *Bifidobacteria*. It turns out that *Lactobacillus* species protect the body from certain infections like *Salmonellosis*, and *Bifidobacterium* species prevent yeasts or bacterial pathogens from growing too much in the intestinal wall. This may reduce the risk of colon cancer by outcompeting putrefactive microbes that can release agents that cause cancer.

Historical records indicate that honey has long been used as a traditional treatment for microbial diseases [25]. Manuka (*L. scoparium*) honey has been documented to be beneficial against a variety of human infections, including *Enterobacter aerogenes*, *Staphylococcus aureus*, *Salmonella typhimurium*, and *Escherichia coli* [21].

Honey has been shown in several tests to be highly efficient against streptococci, vancomycin-resistant enterococci, and methicillin-resistant *S. aureus* (MRSA) [3]. Nevertheless, due to their greater selectivity against medically significant organisms and their enhanced antibacterial potential for local production (hence accessibility), research has recently discovered that bee honeys may be superior to or similar to manuka honey [20].

### 2.2. Immunomodulatory and Anti-Inflammatory Properties

Although inflammation is an essential component of the body’s normal reaction to infection or damaged tissues, excessive or delayed inflammation can impede recovery or even worsen the situation.

The generation of free radicals within the tissue is the most dangerous consequence of mild inflammation. Certain leucocytes that are stimulated as a major part of the inflammatory process are responsible for initiating these unfastened radicals. Inflammatory processes set off a chain reaction of cellular events that lead to the initiation of growth factors that affect the proliferation of fibroblasts, angiogenesis, and epithelial cells [26].

A number of honey varieties from various nations, including those made by stingless bees, have been claimed to have anti-inflammatory properties [13]. Bee honey’s anti-inflammatory properties are a result of its high content of phenolic components. These phenolic and flavonoid chemicals promote the inhibition of the pro-inflammatory effects of cyclooxygenase-1 and cyclooxygenase-2 (COX-1 and COX-2) and/or inducible nitric oxide synthase (iNOS) [27].

Furthermore, the content of prostaglandins, such as prostaglandin E2 (PGE2), thromboxane B2 (in the plasma of healthy individuals), and prostaglandin F2α (PGF2α), decreases when diluted raw bee honey is consumed [15]. There are findings on how various forms of honey can induce the production of interleukin 1β (IL-1β), IL-6, and tumor necrosis factor-alpha (TNF-α) [28,29].

It has also been demonstrated that various honeys, including Gelam honey, lessen inflammatory response mediators like TNF-α and COX-2 by impairing Nuclear Factor kappa B (NF-κB) translocation to the nucleus and blocking the NF-κB pathway. It is commonly recognized that NF-κB activation plays a crucial role in the pathophysiology of inflammation. The sluggish absorption of honey is thought to be the cause of the synthesis of fermentation agents such short-chain fatty acid (SCFA), which has been shown to have immunomodulatory properties. This indicates that some honey-derived fermentable sugars, including nigero-oligosaccharides, have the capacity to trigger an immunological reaction [19,30].

Furthermore, honey contains non-sugar compounds that may be in charge of immunomodulation [31]. Similarly, the topical administration of honey has been discovered to reduce the amount of edema and exudate in wounds, both of which are linked to the activity of the local inflammatory process in wounds [32,33].

### 2.3. Antioxidant Properties

The antioxidant activity of bee honey in the human body is attributed to its capability to reduce oxidative processes, as measured by its capacity to scavenge free radicals [34]. Honey’s anti-inflammatory properties might at least be partially attributed to its antioxidant action, as oxygen free radicals are implicated in a number of inflammatory components [29].

Various flavonoids (such as chrysin, pinocembrin, quercetin, apigenin, and galangin), ascorbic acid, Maillard reaction products, and peptides are found naturally in honey.

Moreover, in honey we can also find tocopherols, catalase, superoxide dismutase, reduced glutathione, and phenolic acids (such as ferulic, ellagic, caffeic, and p-coumaric acids), the majority of which act in concert to provide a synergistic antioxidant action [35,36].

Honey’s antioxidant effect is achieved by inhibiting the production of free radicals, a process that is typically aided by metal ions such as iron, copper, etc. Some typical components of honey, such as flavonoids and other similar polyphenols, may be able to capture these metal ions in complexes, preventing the initial generation of free radicals [29].

## 3. Honey Composition

About 200 different chemicals may be found in honey, the majority of which are sugars, water, and other components including proteins (enzymes) [37].

The substances extracted from honey harvesting include organic acids, minerals (iron, copper, magnesium, calcium, potassium, phosphorus, zinc, and sodium), vitamins (particularly B6, thiamine, niacin, riboflavin, and pantothenic acid), pigments, phenolic compounds, and a wide range of volatile compounds [38,39,40].

### 3.1. Sugars

About 75% of the sugars in honey are monosaccharides, with the remaining 10–15% being disaccharides and trace quantities of other sugars. The attributes of honey, including its energy value, viscosity, hygroscopicity, and granulation, are attributed to its sugar content [41]. The primary factors influencing a honey’s sugar content are its geographical and botanical origin, climate, processing, and storage [42,43].

The ratio between the concentrations of fructose and glucose can be used as an indication to help categorize monofloral honeys [44]. Fructose makes up the majority of the carbohydrates in practically all varieties of honey, with the exception of some, such dandelion (*Taraxacum officinale*) and rape (*Brassica napus*), whereas the proportion of glucose may be larger than the fraction of fructose; as a result, these honeys typically crystallize quickly [42].

Honey’s sugars and other ingredients can alter when it is being stored. Pentoses and hexoses break down into unfavorable compounds like furans when honey is cooked or kept for an extended period of time. This process involves a gradual enolization and a quick β-elimination of three molecules of water [45]. Furfural, which comes from pentoses, and 5-hydroxymethylfurfural (5-HMF), which comes from hexoses like glucose and fructose, are the two primary furans that are produced [46]. These are the principal by-products of sugar degradation, and the presence of these compounds in food is typically associated with non-enzymatic browning events, such as caramelization, the Maillard reaction, and sugar breakdown in an acidic media. Actually, these furans have been employed as indicators for the heat treatment of food [46]. In addition to the compounds listed above, heat in the presence of amino acids can also produce other products of sugar degradation, such as 2-acetylfuran [47], isomaltol [48], 3,5-di hydroxy-2-methyl-5,6-diidropiran-4-one, and maltol. These products can alter the color, taste, and odor of honey.

### 3.2. Proteins

Depending on the species of honeybee, honey shows a different protein composition. One percent (*w*/*w*) of honey’s ingredients are amino acids, and the amounts of these components vary depending on the honey’s source (nectar vs. honeydew) [49].

According to study [50], the protein content of *Apis cerana* and *Apis mellifera* honey ranges from 0.1% to 3.3% and 0.2% to 1.6%, respectively. Pollen is the primary source of protein in honeys; however, proteins and amino acids can also be found in animal and plant sources, such as fluids and nectar secretions from the salivary glands and pharynx of honeybees [37,51].

Proline is the most prevalent amino acid found in pollen and honey [52], accounting for between 50 and 85% of the amino acids in honey [52,53].

Proline has been used as a criterion to assess honey’s maturity and, in certain situations, to detect sugar adulteration. For pure honey, the recommended limit value for proline is 180 mg kg^−1^ at the very least [49,54].

Honey contains the enzyme glucose oxidase. After being hydrolyzed to produce gluconic acid, it transforms glucose into D-gluconolactone. In addition to gluconolactone, hydrogen peroxide—which has bactericidal properties, as well as others—is produced by glucose oxidase [55].

Because of the interaction between the free amino groups and the carboxylic group on the reducing end of sugars, storage and processing conditions may result in the creation of unwanted compounds between proteins and amino acids (Maillard reaction).

After six months of storage, the concentration of proline fell, and after twelve months, the concentration of aspartic acid reduced. Between 6 and 24 months, there was no change in the concentration of β-alanine. This finding could indicate that, although β-alanine participates in the Maillard process, it is less reactive than other amino acids [56].

### 3.3. Organic Acids

Honey has an approximate organic acid content of 0.5% of its fresh weight [57]. Despite being few in number, organic acids are crucial to the physical, chemical, and organoleptic characteristics of honey [58]. Additionally, organic acids can be employed as markers to determine the genuineness and freshness of honey [59]. According to reports, organic acids support honey’s antioxidant and antibacterial properties [60].

These organic acids are generated from sugars by enzymes released by bees either directly from nectar or during the process of turning it into honey [61]. These acids have an impact on honey’s color, flavor, and chemical characteristics [57].

Honey contains a variety of organic acids depending on the geographical area, including aspartic acid, acetic, formic, butyric, citric, fumaric, galacturonic, gluconic, glutamic, and glutaric acid, lactic, malic, malonic, methylmalonic, 2-oxopentanoic, propionic, pyruvic, quinic, shikimic, succinic, tartaric, oxalic, and other acids [57,61,62].

Gluconic acid is the main acid found in honey. The source of this compound is glucose oxidase, which bees supply to honey throughout the ripening process [63]. Apart from gluconic acid, since honey also contains citric acid, the citric acid content of honey can be utilized as a trustworthy indicator to distinguish floral honey from honeydew [57].

### 3.4. Vitamins

Tiny levels of vitamins, particularly the B complex, which are derived from suspended pollen grains, can be found in honey. Thiamine (B1), riboflavin (B2), nicotinic acid (B3), pantothenic acid (B5), pyridoxine (B6), biotin (B8 or H), and folic acid (B9) are among the vitamins that can be found in honey.

Vitamin C, which has mostly been researched for its antioxidant qualities, is present in almost all types of honey. Vitamin C is regarded as an unstable indication due to its high susceptibility to chemical and enzymatic oxidation and its rapid changes in reaction to light, oxygen, and heat [64].

### 3.5. Minerals

Numerous chemical component groups have been identified in many varieties of honey. Some minerals, including potassium, magnesium, and calcium as well as macro- and microelements, silver, zinc, lithium, cobalt, nickel, cadmium, copper, barium, chromium, selenium, sodium, manganese, iodine, zinc, iron, and phosphorus are present in various honeys [38], making up those groups that are significant for the human diet [65].

According to Alqarni et al. [38], the mineral concentration of honey varies from 0.04% in light honeys to 0.2% in dark honeys.

The amount of trace elements in honey changes depending on the type of soil the plant and its nectar were found in, because honeybees’ collection of plants’ chemical components is reflected in the honey [37,65].

When Nalda and colleagues [66] identified minerals in several Spanish honeys, they discovered that honey made from heather and ling flowers had a greater manganese content. Ajtony and colleagues [66] assessed the mineral composition of honey samples from Hungary and discovered that linden honey had the lowest concentrations of lead, copper, chromium, and cadmium, while these elements were present in higher concentrations in other varieties of honey.

The most prevalent element in honeys is normally potassium, which makes up almost one-third of all the minerals in honey [38].

Mineral elements are not degraded by exposure to heat, light, oxidizing agents, high pH, or other circumstances that damage organic nutrients.

### 3.6. Polyphenols

Polyphenols represent a diverse group of chemical substances that can be further classified into non-flavonoids (phenolic acids) and flavonoids (flavonols, flavones, flavanols, anthocyanidin, chalcones, and isoflavones). These substances are the result of secondary plant metabolism and are distinguished by the presence of several phenolic groups connected to complicated structures.

Unlike the basic metabolites (chlorophyll, amino acids, and simple carbohydrates), which mediate in the processes of absorption, respiration, transport, and differentiation of plants, the secondary metabolites have significant ecological roles [67].

Honey’s phenolic composition is mostly determined by its floral origin; in fact, this characteristic can be utilized to classify and authenticate honey, particularly in the case of unifloral types.

Honey’s capacity to scavenge free radicals and transform them into less harmful, more stable molecules is what gives it its antioxidant properties. By releasing hydrogen from their hydroxyl groups, phenolic compounds neutralize free radicals; the activity of the chemical is determined by the amount of groups [68].

The majority of the naturally occurring low-molecular-weight flavonoids are soluble in water. They are composed of two benzene rings that are alternated with a linear carbon chain consisting of three atoms (C6-C3-C6). This structure frequently reorganizes to form three rings, A, B, and C, each containing fifteen carbon atoms. When flavonoids are not linked to sugars, they are referred to as aglycones. Flavonoids are divided into groups based on how much the C ring has been oxidized: isoavones, anthocyanins, anthocyanidins, flavonols, flavones, and flavonones. Among them, flavones, flavonols, and flavonols are the most prevalent in honey [69].

A phenolic ring and at least one organic carboxylic acid function are characteristics of phenolic acids, also known as phenol carboxylic acids. These acids are classified into three groups based on their structure: C6-C3 (p-coumaric, ferulic, and caffeic acid), C6-C2 (acetophenones and phenylacetic acids), and C6-C1 (syringic, vanillic, and gallic acid). The majority of these substances are typically attached to cellulose and lignin, which are the plant’s structural components, as well as to other organic molecules like glucose, other sugars, or flavonoids [31].

Honey’s phenolic content varies both qualitatively and quantitatively depending on the source and botanical origin of the honey nectar, and the phenolic compounds present in honey can be a determining factor in the manufacture of certain honeys (like chestnut).

Moreover, phenolic compounds have been linked to honey’s health benefits, such as its antimicrobial, antioxidant, and anti-inflammatory properties. Phenolic compounds can also act as marker compounds to help distinguish between different types of honey.

Numerous studies have found a correlation between honey’s phenolic content and antioxidant activity [18]. The phenolic profile of honey also reveals the type of bees, their botanical origin, the season, and the area. In one study, the phenolic content of honey from stingless bees and common honeybees is compared. In comparison to Tualang honey, which has a total phenolic content of 183 mg GAE (gallic acid equivalent), Malaysian stingless bee honey has a total phenolic content of around 235 mg, while its flavonoid concentration is 100 mg CE (catechin equivalent). The antioxidant content of stingless bee honey was found to be substantially higher than that of regular honey in the same study [70].

The primary medicinal properties of honey are ascribed to its polyphenol content, since these compounds are the most prevalent phytochemicals and have drawn significant interest from scientific communities as potential preventive agents against degenerative and chronic inflammatory diseases [71]. The bioaccessibility and bioavailability of polyphenols in the human body regulate these functions. Polyphenols that are broken down mechanically and biochemically in the small intestine or liberated from the food matrix have the potential to be both bioavailable and bioactive in terms of bioaccessibility.

Nevertheless, bioaccessibility may be altered or increased depending on how polyphenols interact molecularly with other food ingredients throughout the digestive process [72]. The release of polyphenols for gastrointestinal tract absorption is largely dependent on the presence of dietary fibers in the meal matrix [73]. Less than 10% of polyphenols (aglycones and glucoside conjugate) are absorbed in the upper gastrointestinal tract as a result of the influence of food matrix, according to a small number of clinical trials, while the remaining phenolic compounds go through microfloral metabolization, at which point bioactive metabolites cross the colonic mucosa to plasma [74].

It is also evident that polyphenols may have a strong affinity for fiber and protein, particularly when all of the aforementioned elements are present together. It is therefore possible that the presence of polyphenol in a food matrix including minuscule levels of dietary fiber, protein, and fats will decrease molecular interactions and raise the ratio of bioaccessibility to bioavailability from honey.

Regarding this, manuka honey did not change in terms of its phenolic content or antioxidant activity when compared to other commercial honey samples during the in vitro digestion process [75]. Furthermore, post- and pre-digested manuka honey (ranging from 1–3 mg/mL) had a strong protective effect against hydrogen peroxide-induced DNA damage in the Caco-2 cell line. This shows that manuka honey’s polyphenols are not impacted by any molecular alterations brought on by the digestive process, possibly contributing to their high bioaccessibility ratio [75].

The most common polyphenols in honey are shown in Table 1 and then discussed in text.

#### 3.6.1. Apigenin

Apigenin (4,5,7-trihydroxyflavone) is a low-toxicity flavonoid found abundantly in plants and has been extensively explored. Apigenin is mostly found in fruits, vegetables, nuts, herbs, tea, and honey.

The primary sources of this chemical are plants in the Asteraceae family, specifically those in the genera *Artemisia* [89], *Achillea* [90,91], *Matricaria* [92], and *Tanacetum* [93]. Numerous pieces of research conducted over the years have suggested that apigenin possesses a wide range of intriguing pharmacological properties and prospective applications in nutraceuticals. Honey contains approximately 0.03 mg/100 g of this chemical [16].

Some authors have reported that apigenin, by triggering Wnt/b-catenin signaling, promotes osteogenic differentiation of mesenchymal stem cells and quickens the repair of bone fractures [76]. According to Zhang et al., apigenin boosted the expression of Runx2 and Osterix to promote bone formation and improved the osteogenic differentiation of human mesenchymal stem cells via JNK and p38 MAPK pathways [94].

Moreover, apigenin anticancer effects in a variety of cell lines have been extensively documented. It demonstrated an anti-proliferative effect on liver, breast, colon, neuroblastoma, and cervical cancer cell lines [95].

#### 3.6.2. Chrysin

Chrysin (5,7-dihydroxy-2-phenyl-4H-chromen-4-one) is a flavonoid that naturally occurs in the flavone class of polyphenolic chemicals with a 15-carbon structure. Chrysin is mostly found in nature in passion fruit (*Passiflora* sp.), honey, and propolis [96,97,98].

Many varieties of honey have elevated quantities of chrysin. Forest honey has a chrysin concentration of 0.53 mg/100 g [99].

By stimulating the ERK/MAPK pathway, chyrsin can promote osteoblastic differentiation and mesenchymal stem cell proliferation [100]. Moreover, chrysin can stimulate many signaling pathways that are involved in the development of dental plug stem cells into osteogenic or odontogenic forms. Osteogenic differentiation is produced by increased alkaline phosphatase (ALP) activity and modified expression of osteocalcin, Runx2, and collagen type 1 by chrysin [77]. Chrysin causes osteogenic differentiation of dental pulp stem cells through the activation of the Smad3 signaling pathway, a crucial part of the TGF-β signaling system [77].

#### 3.6.3. Galangin

Honey naturally contains the flavonoid galangin (3,5,7-trihydroxy-2-phenylchromen-4-one, also known as 3,5,7-trihydroxyflavone). According to Habryka et al. [101], honey shows a galangin concentration of about 0.334 mg/100 g. High amounts of galangin can be found in honey and *Alpinia officinarum*, a plant that has long been used in Asia as a spice and herbal remedy for a wide range of illnesses. Galangin has several biological properties, such as the ability to modulate metabolic enzymes, and has anti-mutagenic, anti-clastogenic, anti-oxidative, and radical scavenging properties [78].

#### 3.6.4. Genistein

Numerous dietary legumes, including fava beans, lupine, soybeans, green lentils, alfalfa, and peas, naturally contain the chemical genistein, a 7-hydroxyisoflavone. As an anti-inflammatory, antioxidant, anti-Alzheimer, antidiabetic, or anticancer drug, genistein is used to treat a wide range of illnesses. Furthermore, through activating the Nrf2 (Nuclear factor erythroid 2-related factor 2) cellular signaling pathway, genistein can shield cells from oxidative stress and cellular toxicity [79].

#### 3.6.5. Luteolin

Luteolin (3′,4′,5,7-tetrahydroxyflavone) is a flavone found in celery, parsley, green pepper, and chamomile tea. It possesses anti-inflammatory and antioxidant characteristics.

According to Hossen et al., honey contains around 0.28 mg of luteolin per 100 g. Numerous biological actions, including anti-inflammatory, antibacterial, antioxidant, and anticancer properties, are known to be exhibited by luteolin [16].

Through the Wnt-β-catenin-BMP2-STAT3 signaling pathway, the administration of the natural flavone luteolin can stimulate astrocyte differentiation and boost astrogenesis in human pluripotent stem cells (hPSCs). Luteolin also stimulates the Wnt/β-catenin signaling pathway to hasten the osteogenic development of human periodontal ligament cells. In type 1 diabetic cardiomyopathy, luteolin reduces oxidative stress and thereby improves outcomes following heart failure [80]. Lutein can also help the wound healing mechanism since it promotes fibroblast proliferation. Systemic luteolin treatment improves wound healing by reducing inflammation and oxidative stress by inactivating NF-κB and upregulating Nrf2 [81].

#### 3.6.6. Myricetin

Chemically known as 3,5,7-trihydroxy-2-(3,4,5-trihydroxyphenyl) chromen-4-one, myricetin is a naturally occurring polyhydroxy flavonoid substance that is extracted from *Myrica rubra* leaves and bark.

Berries, tea, nuts, fruits, and vegetables are the most prevalent sources of myricetin. The amounts in black fruits range from 14 to 142 mg/kg. The most prevalent flavonoid in black currants is myricetin, and the amount of this flavonoid varies greatly between varieties. In addition, myricetin can be found in honey. The range of myricetin is around 29.2–289.0 μg/100 g of honey [102].

In cancer cell lines, myricetin causes apoptosis through its antiviral, anti-carcinogenic, anti-inflammatory, and antioxidant properties. Additionally, thioredoxin reductase activity in A549 lung cancer cells and matrix metalloproteinase 2 (MMP-2) enzyme activity in colorectal carcinoma cells were both suppressed by myricetin [31].

Moreover, myricetin regulates the BMP-2/Smad and ERK/JNK/MAPK signaling pathways more effectively, which improves the osteogenic differentiation of human periodontal ligament stem cells (hPDLSCs) [82].

#### 3.6.7. Pinobanksin

Pinobanksin (3,5,7-trihydroxy-2-phenyl-chroman-4-one) is a phenyl allyl flavanone that is frequently found in fruits, propolis, honey, bee bread, bee pollen, and bee wax in addition to the heartwood of the *Pinus* genus. Recent research has revealed pinobanksin’s biological actions, including its strong antioxidant, anti-inflammatory, and antimicrobial qualities, marking it as promising pharmaceutical.

Since pinobanksin may also be found in many types of propolis and honey that is sourced from across the globe, it can also be referred to as a dietary flavonoid [83]. Pinobanksin is known for its antioxidant characteristics, working by preventing the peroxidation of low-density lipoprotein and by decreasing alpha-tocopherol radicals through its electron-donor properties.

#### 3.6.8. Pinocembrin

One of the principal flavonoids present in a wide variety of plants, fungi, and bee products, primarily propolis and honey, is pinocembrin (5,7-dihydroxyflavone).

Numerous in vitro and preclinical investigations have demonstrated the pharmacological properties of pinocembrin, including anticancer, neuroprotective, anti-inflammatory, antibacterial, and antioxidant properties [103]. According to Jasicka-Misiak et al. [104], this molecule has a concentration of 1.60–1.85 mg/100 g in honey. Pinocembrin, which has been found in propolis and honey, most likely originated in indigenous plant pollen grains and was carried by bees [105].

Pinocembrin can also enhance the biological function of endothelial progenitor cells (EPCs) in the bone marrow via activating the PI3K-eNOS-NO signaling pathway. As crucial transcription factors in osteoblast formation, Runx2 and Osterix, as well as their enhancers, Dlx5 and Msx2, had higher mRNA expression when exposed to pinocembrin [84].

#### 3.6.9. Quercetin

One natural flavonoid that can be found in food, including honey, is quercetin. According to Yayinie et al. [106], honey has a quercetin concentration of 1.67–5.08 mg/100 g.

Quercetin has numerous positive health impacts, including preventing diseases like osteoporosis, lung cancer, and cardiovascular disease. Through the ERK and p38 signaling pathways, quercetin may promote the in vitro osteogenic differentiation and proliferation of bone marrow stem cells (BMSCs) [85]. Numerous investigations have also revealed that quercetin promotes the differentiation of osteoblasts into BMSCs [107].

Since quercetin is one of the most prevalent compounds in plants, honeybee diets are high in it. It is currently unknown to what degree quercetin consumption will lower the content of ingested poisons, although it can support the detoxifying system in honeybees. In honeybees, consumption of quercetin-sucrose paste (10 mg quercetin/g) might decrease fluvalinate’s acute toxicity and prolong bees’ lifespans when pyrethroids are present [108].

#### 3.6.10. Caffeic Acid

Caffeic acid is a phenolic acid that is mostly found in human diets and is formed through the secondary metabolism of vegetables. As a result of its ability to prevent the growth of bacteria, fungi, and insects, caffeic acid helps plants defend themselves against diseases, predators, and other threats.

Caffeic acid is present in honey, and, in particular, has been found in buckwheat (0.54 mg/kg), multifloral (0.23 mg/kg), and linden (0.15 mg/kg) honeys [109]. Caffeic acid has been shown to exhibit antioxidant, anti-inflammatory, and anticancer activities.

#### 3.6.11. Chlorogenic Acid

Another common phenolic acid of honey is chlorogenic acid (3-(3,4-dihydroxycinnamoyl) quinic acid). Lu et al. [110] report that it is a common ingredient in fruits and vegetables, as well as the second primary component of coffee beans. Chlorogenic acid exhibits anti-inflammatory (through GPR37 activation), antioxidant, and anti-apoptotic activities [86,87,88,111]. This chemical is present in honey at a concentration of roughly 0.17 mg/100 g [112].

## 4. Biological Qualities of Honey

### 4.1. Honey Application

While some modern chemicals have failed to treat wounds, honey is the oldest known wound-healing agent. Its antibacterial, antiviral, anti-inflammatory, and antioxidant properties have led to additional experimental study demonstrating its effectiveness in wound healing. According to research, honey can be used to treat and manage acute wounds as well as mild to moderate superficial and partial-thickness burns [80]. More research is required to support the existing evidence, even though some studies have shown that honey is effective in treating wounds and leg ulcers. In particular, Ranzato and colleagues [5] have demonstrated the role, in the wound-healing effects boosted by honey, of epithelial-to-mesenchymal transition activation in keratinocytes and the pivotal role of hydrogen peroxide. In particular, hydrogen peroxide is able to cross the plasma membrane of a cell via a specific transporter (i.e., aquaporin-3, [22]) and then initiates a Ca^2+^-signaling cascade, starting the wound closure process [7,113].

Strong evidence supports honey’s potential benefits in the management of diabetes mellitus [114]. These findings highlight the potential benefits of utilizing honey or other strong antioxidants in addition to prescription antidiabetic medications to treat diabetes mellitus. Nonetheless, a number of uncertainties remain, particularly in relation to the possibility of managing diabetes mellitus through therapies that address both hyperglycemia and oxidative stress. Additionally, honey may have therapeutic benefits for managing diabetes that go beyond only lowering blood sugar levels; honey may also help treat related metabolic complications [114].

According to recent research, honey shows anticancer properties through a number of different routes [115]. According to studies, honey has anticancer properties by interfering with several cell-signaling pathways, such as those that trigger apoptosis [116,117]. Furthermore, honey may be able to prevent a number of tumor types in animal models. Nevertheless, additional research is required to enhance our comprehension of honey’s beneficial effects on cancer [118].

Flavonoids and phenolics, two types of antioxidants found in honey, may lower the risk of cardiovascular failure [16,119]. Flavonoids’ preventive properties, including their antithrombotic, anti-ischemic, antioxidant, and vasorelaxant properties, lower the risk of coronary heart disease. Numerous studies have demonstrated the potential pharmacological role of certain honey polyphenols in lowering cardiovascular diseases [120]. To further validate these molecules in medical applications, however, in vitro and in vivo research as well as clinical trials should be undertaken.

It has been suggested that numerous gastrointestinal illnesses, including periodontal disease and other oral problems, dyspepsia, and oral rehydration therapy, could benefit from honey [121]. Although a clinical trial of manuka honey therapy to promote *Helicobacter pylori* eradication failed to show an effective treatment, in vitro investigations suggest that honey has bactericidal action against *Helicobacter* [122,123].

The advancement of nanotechnology has made it possible for humans to better utilize the potential of numerous natural substances. However, traditional nanotechnology frequently employs harsh synthesis conditions along with hazardous and environmentally damaging chemical substances. Nevertheless, “green chemistry” methods are transforming the potential for creating cost-effective and ecologically friendly technology, even for tissue engineering. Honey appears to be one of the most promising natural substances with which researchers are implementing this new “green” strategy. Moreover, because 3D honey-based porous scaffolds can replicate the architecture of a native microenvironment and may allow for the controlled release of honey at the site of injury for rapid regeneration, they are gaining more interest in the fields of biomedicine and wound healing [124].

### 4.2. Quality of Honey

Natural bee honey is a product produced by honeybees (*Apis mellifera* L.), which use plant nectar as well as excreta from insects that extract plant juice or secretions from living plant parts. These are then mixed with particular bee secretions, kept in honeycombs, evaporated, and allowed to mature there.

The grade of bee honey should be high enough for human consumption. The following parameters are required in each country: variety determination, water content, pH, insoluble impurities (no more than 0.1 g/100 g), proline (no less than 25 mg/100 g), diastase activity (general, no less than eight on the Schade scale), electrical conductivity, free acidity (general, no more than 50 milli-equivalents acid/1000 g), and water content. Color (pFund value) and total phenolic content (TPC) are additional factors. Standard techniques, such as spectrophotometric, refractometric, titration, and melissopalynological approaches, are employed to evaluate the quality of honey [125].

Importing honey has become necessary as a result of the rise in consumer demand for bee honey in recent years due to its health-promoting qualities. Adding less expensive high-fructose corn syrup (HFCS) is one method of faking bee honey. One of the most commonly faked foodstuffs is bee honey [126]. By using the ^13^C/^12^C isotope ratio, Çinar et al. [127] looked for a way to identify bee honey that has been adulterated with HFCS.

Despite the fact that honey’s quality standards are well-established globally, unfortunately none of them take into account the biological characteristics or health benefits of honey [128].

## 5. Conclusions

As a naturally occurring substance, honey contains important minerals, vitamins, proteins, enzymes, amino acids, and bioactive substances including phenolic compounds, which have fascinating biological qualities both in vitro and in vivo.

For instance, by altering a number of molecular pathways, honey can prevent the growth of cancerous cells and effectively combat microbial infections. It also has protective effects on the neurological, respiratory, and gastrointestinal systems.

Researchers have verified that, when compared to other functional foods, honey polyphenols have a high bioavailability value. Nevertheless, a complete picture of the molecular mechanisms of polyphenols abundant in honey is lacking. To completely understand the genome-wide effects of honey and patterns of global gene expression, protein translation, intracellular signaling pathways, and metabolite formation in response to specific chemicals, more research is therefore still required in nutrigenomics analysis.

In conclusion, investigating how honey affects gene expression profiling, focusing especially on human intervention trials (ideally large-scale randomized placebo-controlled studies), will help to develop effective strategies for mitigating chronic inflammatory diseases and provide valuable insights into the prophylactic and therapeutic uses of honey.

Therefore, to encourage the general public to use this nutritious food, to support a healthy lifestyle, and to prevent some of the most frequent pathologies, a greater understanding of the mechanisms and contributing elements of the honey effect will be essential.

## Figures and Tables

**Figure 1 biotech-13-00047-f001:**
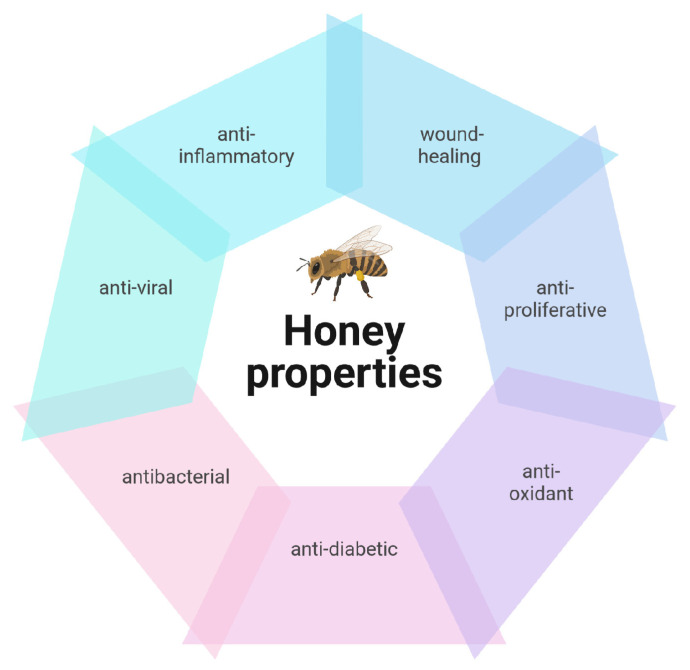
Main recognized properties of honey. For more information, see the text. Created in BioRender. https://BioRender.com/s21r904 (accessed on 11 November 2024).

**Table 1 biotech-13-00047-t001:** The most common polyphenols found in honey, also discussed in the following text.

Compound	Chemical Structure	Mechanism of Action
Apigenin	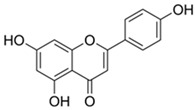	Apigening triggers Wnt/b-catenin signaling, promoting osteogenic differentiation of mesenchymal stem cell. Apigenin also boosted the expression of Runx2 and Osterix to promote bone formation and improved osteogenic differentiation [76].
Chrysin	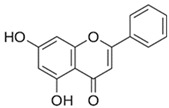	Chyrsin can encourage osteoblast differentiation and mesenchymal stem cell proliferation by activating the ERK/MAPK pathway [77].
Galangin	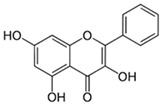	Galangin possesses a number of biological characteristics, including the capacity to alter metabolic enzymes and anti-oxidative, anti-mutagenic, anti-clastogenic, and radical scavenging qualities [78].
Genistein	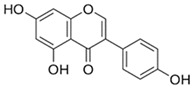	Genistein can protect cells from oxidative stress and cellular damage by triggering the cellular signaling pathway of Nrf2 (Nuclear factor erythroid 2-related factor 2) [79].
Luteolin		Luteolin can promote astrocyte differentiation and increase astrogenesis in human pluripotent stem cells (hPSCs) via the Wnt-β-catenin-BMP2-STAT3 signaling pathway. By deactivating NF-κB and upregulating Nrf2, systemic luteolin therapy enhances wound healing by lowering inflammation and oxidative stress [80,81].
Myricetin	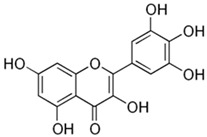	Myricetin enhances the osteogenic development of human periodontal ligament stem cells (hPDLSCs) by more efficiently regulating the BMP-2/Smad and ERK/JNK/MAPK signaling pathways [31,82].
Pinobanksin	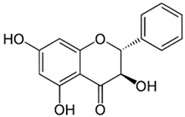	The biological effects of pinobanksin are antioxidant, anti-inflammatory, and antibacterial properties [83].
Pinocembrin	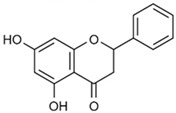	By triggering the PI3K-eNOS-NO signaling pathway, pinocembrin can improve the biological activity of endothelial progenitor cells (EPCs) in the bone marrow [84].
Quercetin	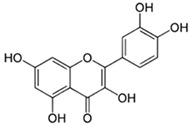	The ERK and p38 signaling pathways allow quercetin to stimulate bone marrow stem cell (BMSC) osteogenic differentiation and proliferation [85].
Caffeic acid	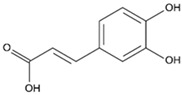	Caffeine has anti-inflammatory, anticancer, and antioxidant properties [86,87].
Chlorogenic acid	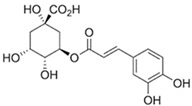	Chlorogenic acid exhibits antioxidant, anti-inflammatory, and anti-apoptotic properties [88].

## Data Availability

No new data were created or analyzed in this study. Data sharing is not applicable to this article.

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
