# Peer review of "Honey Bioactive Molecules: There Is a World Beyond the Sugars"

_biotech, 2024, doi:10.3390/biotech13040047_

Round 1
Reviewer 1 Report
Comments and Suggestions for Authors
1. Abstract: Rewrite the abstract, including the major findings and conclusion.
2. Keywords: - Remove ":", and include "antioxidant, anti-inflammatory; immunomodulatory."
3. Introduction:
- Lines 42-43: “Honey has antibacterial, bacteriostatic, anti-inflammatory, wound healing, antioxidant, radical scavenging, and antiviral properties, making it an effective medicinal agent.”, " this is repeated in lines 23-24.
- Discuss the honey polyphenols.
- Mention the purpose of the task.
4. Table 1 includes bioactive compounds found in honey, along with their mechanisms.
5. Include figure(s) describing honey's antibacterial, antioxidant, and anti-inflammatory properties.
6. Include a section on honey applications.
7. Include a section on determining the quality of honey.
8. The paper should be organized as follows: 1. Introduction, 2. Honey and Humankind, 3. Honey composition, 4. Biological qualities of honey, and 5. Conclusion.
9. A graphical abstract is highly recommended.
Comments on the Quality of English Language- The English language is generally fine but requires minor revisions.
Author Response
Comments and Suggestions for Authors
We appreciate the reviewer's time in assessing our work and making suggestions about how to make the ms better.
- Abstract: Rewrite the abstract, including the major findings and conclusion.
The abstract has been changed.
- Keywords: - Remove ":", and include "antioxidant, anti-inflammatory; immunomodulatory."
We have modified the keywords
- Introduction:
- Lines 42-43: “Honey has antibacterial, bacteriostatic, anti-inflammatory, wound healing, antioxidant, radical scavenging, and antiviral properties, making it an effective medicinal agent.”, " this is repeated in lines 23-24.
We have corrected this mistake
- Discuss the honey polyphenols.
Now, the introduction covers the topic of polyphenols.
- Mention the purpose of the task.
We have outlined the ms purpose.
- Table 1 includes bioactive compounds found in honey, along with their mechanisms.
The primary biological action of honey-derived chemicals has been included.
- Include figure(s) describing honey's antibacterial, antioxidant, and anti-inflammatory properties.
The primary properties of honey are shown in a new figure (figure 1).
- Include a section on honey applications.
A new section about the applications of honey has been included.
- Include a section on determining the quality of honey.
A section on determining the quality of honey has been included.
- The paper should be organized as follows: 1. Introduction, 2. Honey and Humankind, 3. Honey composition, 4. Biological qualities of honey, and 5. Conclusion.
We have rearranged the ms in accordance with the recommendation.
- A graphical abstract is highly recommended.
A graphical abstract has been proposed.
Comments on the Quality of English Language. The English language is generally fine but requires minor revisions.
We have made significant changes to the text. We therefore hope that our ms will be positively accepted.
Reviewer 2 Report
Comments and Suggestions for Authors
The manuscript proposed by the authors provides an overview of honey composition and properties. The topic is well-known and the authors did not focus on new aspects, for example, discussing new studies about biological activities. From this perspective, the manuscript does not contribute significantly to the field. However, the review is well-written and the main aspects are discussed and aligned with the journal. In some sentences, references are lacking.
Author Response
The manuscript proposed by the authors provides an overview of honey composition and properties. The topic is well-known and the authors did not focus on new aspects, for example, discussing new studies about biological activities. From this perspective, the manuscript does not contribute significantly to the field. However, the review is well-written and the main aspects are discussed and aligned with the journal. In some sentences, references are lacking.
We value the reviewer's positive evaluation of the manuscript. We have made significant changes to the text and provided references for some sentences. We therefore hope that our ms will be positively accepted.
Round 2
Reviewer 1 Report
Comments and Suggestions for Authors
Table 1: include relevant references.
Author Response
Table 1: include relevant references.
We value the reviewer's efforts in evaluating our work and offering recommendations for improving the ms. Table 1 now includes pertinent references.